# Microbial Biofilm: A Review on Formation, Infection, Antibiotic Resistance, Control Measures, and Innovative Treatment

**DOI:** 10.3390/microorganisms11061614

**Published:** 2023-06-19

**Authors:** Satish Sharma, James Mohler, Supriya D. Mahajan, Stanley A. Schwartz, Liana Bruggemann, Ravikumar Aalinkeel

**Affiliations:** 1Department of Urology, Jacobs School of Medicine and Biomedical Sciences, University at Buffalo, Buffalo, NY 14260, USA; ss466@buffalo.edu (S.S.); sasimmun@buffalo.edu (S.A.S.); 2Department of Urology, Roswell Park Comprehensive Cancer Center, Buffalo, NY 14203, USA; james.mohler@roswellpark.org; 3Department of Medicine, Division of Allergy, Immunology, and Rheumatology, Jacobs School of Medicine and Biomedical Sciences, University at Buffalo, Buffalo, NY 14203, USA; smahajan@buffalo.edu; 4Department of Medicine, VA Western New York Healthcare System, Buffalo, NY 14215, USA; 5Department of Biomedical Informatics, University at Buffalo, Buffalo, NY 14260, USA; lianabru@buffalo.edu

**Keywords:** biofilm, extracellular polysaccharides, healthcare-associated infection, medical device infections, antibiotic resistance, biofilm control

## Abstract

Biofilm is complex and consists of bacterial colonies that reside in an exopolysaccharide matrix that attaches to foreign surfaces in a living organism. Biofilm frequently leads to nosocomial, chronic infections in clinical settings. Since the bacteria in the biofilm have developed antibiotic resistance, using antibiotics alone to treat infections brought on by biofilm is ineffective. This review provides a succinct summary of the theories behind the composition of, formation of, and drug-resistant infections attributed to biofilm and cutting-edge curative approaches to counteract and treat biofilm. The high frequency of medical device-induced infections due to biofilm warrants the application of innovative technologies to manage the complexities presented by biofilm.

## 1. Introduction

A biofilm is a community of microorganisms, such as bacteria, that are capable of living and reproducing as a collective entity known as a colony. To put it another way, biofilms are living biomass that possess a sophisticated social structure that personnel involved in this field are still attempting to decipher. The structure of biofilm serves both to shield and enable the expansion of the colony.

That a symbiotic relationship exists between prokaryotes and eukaryotes, or unicellular and multicellular organisms, is common knowledge. These symbiotic relationships are mutually beneficial. The human body consists of a microbiome that is large and complex and consists of bacteria, fungi, and viruses. Most of the microbiota in the human body resides in the gastrointestinal tract, salivary mucosa, and skin, where they facilitate various physiological functions ranging from metabolism to innate immunity. However, under certain circumstances, the growth of these symbiotic microorganisms can become uncontrollable and can lead to infections that initiate the formation of biofilms. Since their evolution, bacteria have existed in two separate states: the planktonic state (free-floating) and sessile state (adhered to a surface) [1]. Bacteria exhibit different traits between these two states because bacterial attachment to a surface causes a rapid change in the expression levels of several genes associated with maturation and production of exopolysaccharide (EPS), also known as “slime” or bacterial EPS. A protective barrier is produced because of this transition, which starts immediately after bacteria colonize both biotic and abiotic surfaces [1,2,3]. This barrier shields the bacteria from the natural defense mechanisms of the host and from external threats, such as antibiotics. Anthony van Leeuwenhoek first observed surface-associated bacteria, but the word “biofilm” was not used or defined until a manuscript by Costerton et al. [4]. The significance of biofilms was acknowledged by the American Society for Microbiology in 1993 [4]. Costerton et al. characterized biofilm more fully in 1999 as an organized population of microbes encased in a polymeric matrix produced by the microbe that is adhering to a surface [5]. Biofilms impact all facets of human life, from public health to industrial concerns, and impact the economy, use of energy, equipment degradation, contaminated products, and infections. Cutting-edge technologies such as scanning and confocal microscopy have helped scientists understand the extraordinarily intricate structure of biofilms (Figure 1). The use of these cutting-edge technologies has discovered that biofilms are complex populations of cells wrapped in a matrix of EPS with permeable water channels and uniform deposits of cells and collected slime. Many human diseases and the colonization of medical devices are linked to microorganisms growing in biofilms and these microorganisms are very resistant to antimicrobial treatments. Biofilm formation initiates the disease process through various mechanisms, such as the detachment of individual bacterial cells or clusters of cell aggregates, the production of endotoxins, heightened evasion from host immune system surveillance, and the establishment of a protective barrier conducive to the emergence of immune-resistant organisms.

Modern understanding of this subject defines biofilms as an immobile complex structure comprising of single or multiple species of bacteria, host cells, and cellular by-products, with the cells irreversibly attached to the substratum and surrounded by an extracellular polymeric substance produced by bacteria. A surface that provides moisture and nutrients is the ideal environment for biofilm development. Biofilms can be good, bad, or neutral [6]. Biofilms that are part of a natural environment are neutral, whereas biofilms that grow on open wounds following infection are harmful. Biofilms may play a constructive role in solving ground contamination from an oil spill. Biofilms are responsible for 70% of all microorganism-induced infections and are a significant contributor to healthcare-associated infections (HAIs) in humans. The microorganism living as part of the biofilm shows characteristic features, such as collective cooperation, source capturing, and increased survival against treatment with antimicrobials. Increased survival and evasion of the host immune system make biofilms responsible for persistent chronic infections [7]. The complexity of biofilm activity and behavior requires multidisciplinary research to develop an effective solution against the devastation that can be caused by this structure.

## 2. Makeup of and Nature of Bacteria in the Biofilms

The makeup of biofilm is 10% microbial mass and 90% water [8]. A range of 50–90% of the entirety of the organic component of biofilms is attributed to the polysaccharides that form the matrix [9]. Chains of polysaccharides are woven together in a dense, mesh-like structure [10,11]. The hydroxyl groups on the polysaccharide increase mechanical strength by interacting with each other [10,11]. The biofilm architecture can have positively charged ions, such as Ca^2+^ or Mg^2+^, which form supportive cross bridges between polymers and allow biofilms to grow to thicknesses of up to 300 µm. In other instances, the polysaccharides in biofilms can be neutral or polyanionic, as in the EPS of Gram-negative bacteria [12]. Biofilms also can have uronic acids, such as D-glucuronic, D-galacturonic, and mannuronic acids or ketal-linked pyruvates that bestow anionic properties [5,12]. Anionic properties allow the association of divalent cations to inter-link strands of polymer and provide a greater binding force to matured biofilm [13]. In the case of Gram-positive bacteria, such as *staphylococci*, the chemical composition of EPS is entirely different and is principally cationic. Hussain et al. have reported that the slime of coagulase-negative bacteria consists of a teichoic acid mixed with small quantities of proteins [14]. The different charges and ions in the biofilm provide structural integrity to the EPS, which confer biofilms with the property to withstand environments of extreme shearing forces, such as on waterfall impact points.

Bacteria growing in biofilm are sessile and are responsible for most physiological processes in the biofilm environment [15]. The sessile bacterial biofilm communities have different growth, gene expression, transcription, and translation rates. These functional characteristics are acquired by the sessile bacterial biofilm communities in the process of adaptation to microenvironments that have higher osmolarity, scarcer nutrients, and increased cell density. The resulting structure of a biofilm is extremely viscoelastic and has a rubbery behavior [16]. The most frequently found bacteria in a biofilm are *Pseudomonas aeruginosa*, *Staphylococcus epidermidis*, *Escherichia coli*, *Klebsiella pneumoniae*, *Proteus mirabilis*, *Streptococcus viridans*, *Staphylococcus aureus*, and *Enterococcus faecalis* [17]. *Staphylococci* are a diverse group of Gram-positive bacteria and generally found in the skin and mucosa of mammalians. Bacteria belonging to this genus are responsible for nearly 80% of infections caused by implantable devices in humans [18,19,20,21,22,23,24,25,26]. A recent NIH study reports that 70% of all human microbial infections are due to biofilms and lead to various diseases including non-healing chronic wounds, endocarditis, periodontitis, cystic rhinosinusitis, fibrosis, meningitis, osteomyelitis kidney infections, prosthesis, and implantable device-related infections [2,3,18,27,28,29,30,31]. Extreme care in the manufacturing process seeks to maintain sterility of an implantable device, but a device that becomes contaminated during and after implantation that can cause serious device-associated infections that require removal and can result in death.

## 3. Formation of Biofilms

The formation of the three-dimensional architecture of biofilm is a several step process and involves adsorption, adhesion, microcolony formation, maturation, and dispersion (Figure 2). A biofilm surface’s solid–liquid intersection with an aqueous medium (such as blood or water) offers the perfect condition for microbe attachment and growth. The close association of the cells in the biofilm colony creates the conditions encouraging the development of gradient in nutrition availability, exchange of genes, and quorum sensing (QS).

Biofilms are formed in a turbulent flow environment where the Reynolds number (Re) is more than 5000. Re, a non—dimensional number used in fluid mechanics, measures the proportion of inertial to viscous forces and aids in the prediction of fluid flow patterns in various contexts. A lower Re number suggests laminar flow and a higher Re number reflects turbulent flow. Turbulent flow enhances biofilm formation. Smooth and rough surfaces have been reported as being colonized with equal ease; a biofilm surface’s physical attributes affect bacterial adhesion only to a marginal level [32]. Studies performed to comprehend the mechanism of biofilm development in the 1970s were limited by the available research tools. The advent of advanced instrumentation and technologies has revealed that biofilms are adaptive structures employed by microorganisms as a defensive shield to create an advantageous environment that helps them retain nutrients and guarantee survival in an unfavorable environment [33]. Further, biofilms produced by different microorganisms have a great degree of similarity, but they also can have minor traits that are unique to a particular species [3,18,29,34]. Studies performed in the early 21st century describe how biofilm formation is governed by natural forces and physiological events [3]. The establishment and development of biofilm is characterized by five stages: (i) initial attachment (reversible and irreversible) of single bacteria, (ii) bacterial aggregation (iii) microcolony formation, (iv) maturation, and (v) dispersion/detachment [35,36,37,38,39]. Production of bacterial adhesins, which aid in adhesion to surfaces, is a hallmark of the first phase. The initial colonization of any surface is performed by free floating planktonic bacteria. These free floaters attach to the surface, multiply, become sessile, acquire various additional characteristics based on the environment, and attach covalently to the surface. 

### 3.1. Initial Attachment

Attachment begins when the free-floating planktonic bacteria encounter any surface. They cling to the surface via bacterial appendages, such as flagella and or pili, or by physical forces [3,28,29]. The process is more likely a chance encounter and the attachment at this stage is transient and easily reversible [29,31]. The degree of adherence of the bacteria to the encountered surface is governed by a wide range of circumstances, including material composition, surface properties of the bacteria cell, temperature, and pressure [31]. The forces that control the degree of attachment may include hydrophobic, steric, electrostatic, van der Waals, and protein adhesion. The cumulative effect of these forces helps the bacteria persist in sticking to the surface and overcome the forces of repulsion such that they become irreversibly attached to the surface and form a monolayer [29,31,40].

### 3.2. Bacterial Adhesion and Aggregation

The second stage of adherence is referred to as the anchoring or latching phase, and it involves a binding that is molecularly coordinated among particular adhesins and the outermost layer [41]. At this stage, the loosely connected organisms solidify by the adhesion process by creating EPS that interact with surface materials and/or receptor-specific ligands present on pili, fimbriae, and fibrillae, or both. This helps the organisms adhere more securely to the surface they are attached to. On culmination of the second stage, the adhesion will have become irreversible in the absence of any physical or chemical intervention, and the organism will be aggregated on the surface in an irreversible stable way, similar to how a cocoon would attach to a leaf. Depending on the conditions present, particular organisms may make use of a variety of different adhesins in order to adhere themselves to surfaces. At this point in the process of adhesion, planktonic microbes are capable of adhering to one another as well as to various kinds of surface-bound organisms, resulting in the formation of aggregation on the substratum. It is interesting to note that the existence of a particular kind of microorganism on an outer layer might stimulate the adherence of another species of microorganism [42,43]. Each bacterium produces various adhesins, and some of these adhesins are controlled at the transcriptional level. This allows organisms to transition from a sessile to a planktonic form depending on the environmental factors that they are exposed to [42,44]. This is the case with S. epidermidis that generates a type of polysaccharide intercellular adhesin (PIA) which is important for cell-to-cell adhesion and, as a result, the formation of biofilm [45,46,47,48].

### 3.3. Microcoloy Formation

The next stage after bacterial attachment is multiplication and cell division to form microcolonies. This process is triggered by specific chemical signaling within the EPS and micro communities [5,49]. In a biofilm, bacterial colonies typically contain a variety of micro-communities. In many ways, these micro-communities cooperate with one another. This cooperation is essential for substrate exchange, the flow of significant metabolic products, and the elimination of metabolic waste. For example, a minimum of three different species of bacteria must be present for anaerobic digestion to occur and for complex organic matter to be broken down into CH_4_ and CO_2_. Following the breakdown of complex organic compounds, bacteria that ferment begin to generate acid and alcohol from organic compounds. Acetogenic bacteria then consume these substrates as their food, and methanogens obtain energy by converting acetate, carbon dioxide, and hydrogen into methane. Biofilm offers a comprehensive environment for the establishment of syntrophic association. Syntrophic association is the affiliation of two or more metabolically distinct bacteria that rely on one another to utilize specific substrates for their energy needs [50].

### 3.4. Maturation

Stage four of biofilm formation is maturation where the attached cells mature and develop further. Maturation is enabled by the secretion of signaling factors by the attached bacterial cells resulting in the expression of biofilm-specific genes. Signaling factors help alter gene regulation to enhance bacterial virulence. The process begins with the release of EPS from the cells, which stabilizes the biofilm structure and shields it from antimicrobial agents [3,29]. For example, *P. aeruginosa* generates unique saccharides (alginate, Pel, and Psl) throughout maturation that offer biofilm stability [3]. According to a report, environmental DNA (e-DNA) is responsible for intracellular signaling and biofilm strength [3]. Besides aiding initial attachment, the *S. epidermidis* polysaccharide intercellular adhesion (PIA) antigen shields the proliferating bacteria against polymorphonuclear leukocytes [51]. Many layers of cell clusters are created during their accumulation and aggregation on the surface. These clusters eventually develop into microcolonies, which are likewise encased within the EPS, where quorum sensing (QS) and intercellular signaling occur. Overall, maturation occurs in two stages: Stage I entails cell-to-cell contact and the synthesis of autoinducer signal molecules, such as N-acylated homoserine lactone (AHL), whereas stage II comprises an expansion of the microcolony size and thickness to about 100 µm, which is the definition of an established microcolony [3,18]. The connections made by the bacteria in the biofilm are facilitated by active collaborations, and the connectivity between them depends on how far apart they are from one another [36]. During the maturation stage, bacteria are able to recognize the dimensions and proximity of adjoining groups, which aids them in forming clusters that can bond with nearby cells more effectively [36]. Gene and protein expression is regulated by the complete bacterial colony in the biofilm rather than through individual bacterial cells [36]. In conclusion, the second stage entails EPS generation, cell aggregation, chemical bonding, QS, and creation of micro- and macro-colonies [3,40].

### 3.5. Dispersion

Dispersion, the process by which bacteria spread from one part of an infected person’s body to another to spread infection, is an important phase in the growth of biofilm development. A biofilm typically has two clearly separated layers [29]. One component is the foundational layer where the bacteria primarily reside, whereas another layer, known as the surface layer, functions as a dispersal zone where they disseminate into their surroundings, facilitating their spread and sustained presence. Chronic infection and other severe conditions such as embolic problems are brought on by this stage and necessitate rapid medical attention [29]. As a result, metastatic seeding is a common name for this mechanism [1,3,30,40]. Resources become scarce as the biofilm ages, and metabolic byproducts that are toxic build-up. The microbial cells, therefore, scatter to other parts of the infected host or other areas of the medical implant to grow, obtain nutrients, and expel stress-inducing situations and waste byproducts [3,29,52]. Single cells or aggregates of cells that are peeled from the biofilm make up to initiate the dispersion process [3]. This process is believed by some researchers to be a programmed process that is triggered by either nutritional deficiency or oxygen level in the case of aerobic bacteria. In response to the deficiency created, tiny molecules such as the fatty acid DSF (cis-11-methyl-2-dodecenoic acid) are activated by autophosphorylation. This autophosphorylation causes the cyclic diguanylate Guanosine monophosphate (c-di-GMP) phosphodiesterase to become activated, which breaks down c-di-GMP. When c-di-GMP is degraded, clusters of the biofilm are torn apart by shear pressures or discharge planktonic cells which then dissolve parts of EPS [27,29,52]. In addition to gene regulation pathways, there are additional mechanisms involved in the breakdown of EPS. These mechanisms include the release of enzymes by bacterial cells that aid in the lysis of saccharides. This activity dissolves the polysaccharide matrix anchoring the biofilm, thereby freeing the top layer of bacteria [29,52,53]. Once liberated, the bacteria either create new biofilms in different organ of the body or float at liberty on the surface by stimulating the production of proteins that aid in motility [3,29]. 

### 3.6. Quorum Sensing

In reaction to the density of the population’s cells, quorum sensing (QS), a method of cell–cell interaction, synchronizes gene expression. The processes of QS and biofilm formation are interdependent. When the QS gene is activated, it causes the biofilm to develop and then coordinates its maturation and breakdown. The phenomenon of QS is feasible only when a specific volume has a minimal quantity of bacteria. The amount of autoinducer signaling molecules secreted by the bacteria in a microcolony can be used to determine the number of bacteria in each volume [34,36,54,55]. Some researchers, however, disagree, arguing that the autoinducer signaling molecules are simply metabolic byproducts and not actual signaling molecules, and hence they should not be considered signaling molecules [56]. To comprehend the QS process used by bacteria when building biofilms, various mathematical and modeling approaches have been presented. Because QS is critical for the development and maturation of biofilms, researchers believe that inhibiting this process could be a valuable weapon in the battle against biofilms. As a result, a new area of research called “quorum quenching” is being developed to find products and substances that can “quorum quench.” Last but not least, research on *P. aeruginosa* and *B. cepacian*, two distinct species of bacteria linked with biofilms, has revealed that quorum quenching products reduce antibiotic resistance in bacteria [57].

## 4. Infections Attributed to Biofilms

Several studies on infectious diseases have pointed to an association with biofilms [5,58,59,60,61,62]. Conservative estimates suggest that roughly 70% of all bacterial infections are linked with biofilm and are both device-related and non-device-related [63]. Non-device related infections occur because one’s own body provides a suitable biotic surface having ideal moisture and other support systems for bacteria to attach and produce EPS. For instance, *P. aerobicus* and *Fusobacterium nucleatum* is the cause for periodontitis when oral cleanliness is inadequate by infecting the gingiva [64]. The biofilms that develops on the surface of the tooth intervene with the transit of calcium in the epithelial cells, resulting in the formation a mineralized biofilm (plaque or tartar) predominantly composed of calcium and phosphate ions [65]. Another biofilm-related, non-device illness that is transferred through the bloodstream to the bone metaphysis is osteomyelitis [66]. Immune system actions to the microbe cause the bone tissue to degenerate even more and shatter [66]. Furthermore, the biofilms that grow in the open wounds of diabetic patients is believed to cause chronic illness [67]. As aerobic bacteria create biofilms on the exterior of deep wounds, anaerobic bacteria invade the interior [68,69]. Numerous biofilm-associated microorganisms are responsible for conditions such as otitis media chronic prostatitis, native valve endocarditis, cystic fibrosis, and periodontitis [70]. Biofilms can lead to infectious disease in the following ways: (a) detachment of biofilm cells or masses of cells leading to the blood and urine infections or emboli formation, (b) cells can interchange resistance plasmids within biofilms, (c) decreased susceptibility of the cells to antimicrobial agents, (d) endotoxin production by the biofilm-associated by bacteria, and (e) resistance host immune system. 

## 5. Biofilms on Medical Devices

Costerton et al. was the first to demonstrate an association between biofilm and medical devices [5]. Subsequent studies have demonstrated urinary catheters, central venous catheters, indwelling stents, contact lenses, intrauterine devices, and dental chair water lines all to be susceptible to bacterial adhesion and biofilm formation [71,72,73]. Catheters are inserted to administer liquids, blood and blood products, drugs, nourishment, and for hemodynamic monitoring [74]. The outer or inner lumen of the catheters can have biofilms. Routes for bacterial transmission include ascent up the catheter’s outside surface or transmission through the main channel. Platelets and other tissue proteins form the conditioning films on the surface of catheters [74,75]. Adhesins included polysaccharide intercellular adhesin and hemagglutinin [76]. 

### 5.1. Prosthetic Heart Valves

Karchmer and Gibbons demonstrated biofilm’s association with prosthetic heart valves [77]. Prosthetic heart valves may be of two types: mechanical valves and bio-prostheses or tissue valves. However, the infection rates are similar in both [78]. Adjacent tissue damage during surgical implantation may lead to accumulation of platelets and fibrin with the potential for microbial colonization. Infection associated with prosthetic heart valves is known as prosthetic valve endocarditis (PVE) [79,80,81,82]. PVE is classified as early (≤12 months) or late (>12 months) after operation. The microbial organisms responsible for PVEs are predictable based on the time after valve implantation. Time of infection may reflect the pathogenic mechanism [82,83,84,85,86,87,88,89]. In the first 2 months after valve implantation, the most common pathogens are coagulase-negative staphylococci (CoNS) and *S. aureus*, followed by members of the *Candida* species and by Gram-negative bacilli. This range of bacteria reflects the usual nosocomial origin of these infections. From 2 to 12 months after valve implantation, the extremely common pathogens are coagulase-negative *Staphylococci*, *S. aureus*, and *Streptococci*, followed by *Enterococci*. After a year following the installation of a valve, the typical bacteria are CoNS, *S. aureus Streptococci* and *Enterococci*. 

### 5.2. Central Venous Catheters

Maki et al. was the first to demonstrate that central venous catheters (CVCs) are more vulnerable to device-related infections than any other indwelling medical device [74]. Three days after catheterization is the normal timeframe for colonization and biofilm growth on CVC. Raad et al. showed that catheters left in place for less than 10 days tended to generate biofilm more on the outside of the catheter, whereas catheters left in place for 30 days or more tended to form biofilm more extensively and frequently on the inside of the catheter [90]. Pathogens colonizing CVCs include *S. aureus*, *P. aeruginosa*, *Klebsiella pneumoniae*, *Enterococcus faecalis*, and *Candida albicans* [91]. The distal tip of the catheter is withdrawn aseptically and rolled over the surface of a nonselective media to identify CVC biofilms. The amount of organisms recovered upon contact with the agar surface determines the size of the biofilm on the catheter tip [75]. The roll-plate method for diagnosing catheter-related bacteremia has poor detection accuracy and poor predictive value, according to Slobbe et al. [92]. They observed that even a threshold of 104 CFU/tip indicated catheter-related septicemia by sonicating and vortexing catheter tips to improve biofilm quantification.

### 5.3. Contact Lenses

Contact lenses are hard or soft depending on the manufacturing material. Soft contact lenses composed of hydrogel or silicone allow oxygen diffusion through the lens material to the cornea. Hard contact lenses made of polymethylmethacrylate allow oxygen-containing tears to flow underneath the lens through movement with every blink. Both types of lenses are easily colonized by bacteria [93]. *P. aeruginosa*, *S. aureus*, *S. epidermidis*, *Serratia spp*., *E. coli*, *Proteus* spp., and *Candida* spp. are bacteria that have been reported to cling to contact lenses [94]. Miller and Ahearn concluded that the rate of adherence of *P. aeruginosa* to hydrophilic contact lenses varied based on water content [95]. Type of bacteria, pH, and substrate can affect the level of pathogen attachment. Moreover, biofilms develop on contact lens storage cases, and it was found that 80% of lens users without symptoms had contaminated storage cases [96].

### 5.4. Intrauterine Devices (IUDs)

IUDs are made of polyethylene, a nonabsorbable polymer impregnated with barium sulfate. There are also varieties that release chemicals, including copper or a pro-gestational agent. IUD can cause pelvic inflammatory disease [97]. It has been shown that *S. aureus*, beta-hemolytic *Streptococci, E. coli*, and other anaerobic bacteria can be found in IUDs that have been removed from women with pelvic inflammatory disease. On the other hand, it has been observed that IUDs removed from asymptomatic women were highly infected with *S. epidermidis*, enterococci, and anaerobic lactobacilli [98]. Other pathogens that have been identified include *Lactobacillus plantarum, S. epidermidis, Corynebacterium* spp., *Micrococcus* spp., *Candida albicans*, *S. aureus*, and *Enterococcus* spp. In IUD-associated biofilms, Marrie and Costerton demonstrated the existence of human leukocytes and cellular detritus [99]. A potential major source of contamination could be the IUD’s tail. Studies have shown that microcolony production was most prevalent in the distal parts of the tail, which are exposed to the vaginal microbiota [100]. Tatum et al. proposed that the tail of the IUD provides the surface that allows microorganisms to reach the endometrial cavity through capillary action [98,101].

### 5.5. Dental Unit Water Lines

Pathogenic organisms in dental unit water lines can infect patients and dentists [102]. The water for various hand pieces, including the air-water syringe, the ultrasonic scaler, and the high-speed hand piece, is delivered to dental units via small-bore flexible plastic tubing. Water sources could be metropolitan, distilled, or sterile water reservoirs. Furuhashi and Miyamae demonstrated that the bacterial counts had increased from the typical municipal water supply of less than 40 cfu/mL to between 103 and 105 cfu/mL in water samples gathered from the three-way syringe [103]. They also observed that the cup water filler and air turbine hand piece both had high numbers. Whitehouse et al. showed a polysaccharide matrix embedded with a variety of bacteria [104]. Water counts and biofilm were shown to be positively correlated. They reported that even after 180 days of exposure, a dense, multi-layered extracellular polymeric material had completely covered the surface of the dental unit water line. It has also been demonstrated that dental suction devices such as saliva ejectors support biofilms containing both mixed skin microbiota and aquatic microorganisms. 

### 5.6. Urinary Catheters

Urinary catheters are of two types, either latex or silicone devices. They are inserted via the urethra into the bladder to determine urine yield, collect urine during surgery, control urinary incontinence, or relieve urinary retention. The devices are either open or closed. The catheter empties into an open collection receptacle in open devices, which are primarily utilized only in developing nations, as opposed to a plastic collection bag in closed systems used elsewhere. Urinary catheters in the initial stages are colonized by a single species, such as *Enterococcus faecalis*, *E. coli*, *S.epidermidis*, or *Proteus mirabilis*. Subsequently, mixed communities develop, containing organisms such as *Providencia stuartii*, *P. aeruginosa*, *Proteus mirabilis*, and *Klebsiella pneumoniae* [105,106]. Since certain constituent organisms in biofilms on urinary catheters have the potential to change the local pH by producing urease, which hydrolyzes urea in urine to produce free ammonia, these biofilms are distinctive. As a result of the ammonia raising of the local pH, minerals including calcium phosphate (hydroxyapatite) and magnesium ammonium phosphate might precipitate (struvite) (Figure 3). In the catheter biofilms, these minerals will be deposited and form a mineral encrustation [105,106].

The consequence of biofilms on urinary catheters is that patients develop UTI within 4 days of insertion [107]. Virtually all healthcare-associated UTI are due to insertion of a urinary catheter and these UTIs are referred to as catheter-associated urinary tract infections (CAUTIs) [108]. The frequency of CAUTIs are lower in closed systems and urine can remain sterile for 10–14 days in approximately half the patients [109]. Stickler et al. in their study demonstrated that 10–50% of patients undergoing short-term (<7 days) catheterization develop CAUTI, whereas essentially all patients undergoing long-term (>28 days) catheterization develop CAUTI [106]. According to McLean et al., bacterial climbing from the catheter to the bladder is the main cause of the 10% rise in CAUTI risk for each day the catheter stays in place [110]. Nickel et al. stated that biofilms formed in CAUTI have multiple species of bacteria within them resulting in a thick coherent biofilm that confers significant resistance to antibiotics even though individual bacteria in the biofilm remain sensitive, thus accounting for the failure of antibiotic therapy [111]. Moreover, they noted that there was no connection between the length of catheter use and the degree of biofilm growth. The attributable cost of CAUTI exceeds USD 1000 per patient, but the total economic loss annually is about USD 1.7 billion [112]. In 2008, Center for Medicare Services (CMS) listed CAUTI as one of the 14 hospital-acquired and preventable conditions and therefore stopped reimbursing hospitals as part of the Hospital Acquired Conditions Reduction Program (HACRP), creating a financial incentive for CAUTI prevention efforts. 

Common symptoms of CAUTI include increased urinary frequency and urgency, dysuria, abdominal pain, and tachycardia. Catheter obstruction, hematuria, and cloudy urine are signs of CAUTI. The most common species responsible for CAUTI is the Gram-negative bacteria *E. coli*, but *Pseudomonas*, *Klebsiella*, *Proteus* genus, and Gram-positive bacteria such as *Staphylococcus aureus*, *Enterococcus faecalis* have been reported. When indwelling catheters are inserted or the collection system is handled improperly, bacteria from the patient’s colonic or perineal microbiota, from the hands of medical professionals, or from the hands of the patient can enter the urinary tract. Bactria are protected inside the biofilm since it acts as a reservoir of infection and promotes anti-microbial resistance.

CAUTIs are most frequently caused by rectal microbiota contaminating the urethra. Then, the bacteria migrate to the bladder, adhere, and colonize there [113]. Furthermore, bacterial proteases and toxins cause damage to the epithelium. Bacteria then grow and create biofilms (Figure 3). Both with and without a urinary catheter, the basic stages of infection proceed in the same way. With urinary catheters, the bladder can be directly connected to the outside world. Although in certain people this conduit is crucial for urine evacuation, it also serves as a pathway for rectal and periurethral microbe ascent to the bladder, where they can build a base for infection. The risk of UTI is raised by catheters because they bypass the urethral sphincters, lessen the turbulence that normally occurs during spontaneous urination, and act as an infection nidus. Moreover, catheters could irritate and traumatize the uroepithelium, rupturing the mucopolysaccharide layer that normally protects it, and making it vulnerable to bacterial adhesion and invasion. An ideal environment for adhesion by uropathogens that produce fibrinogen-binding proteins is created by the robust immunological response to catheterization, which causes fibrinogen to accumulate on the catheter [113,114]. For instance, *Enterococcus faecalis* does not grow in urine or bind to catheter material in a culture setting, but it does grow in urine that has been supplemented with fibrinogen and clings to a catheter coated with fibrinogen [115]. One important first step in UTI is adherence. Two species of bacteria may stick to the bladder’s uroepithelium in cases of uncomplicated UTI, giving the infection time to take hold. The production of a biofilm is thus enabled by bacterial adhesion to a suprapubic tube or urethral catheter [116]. 

Using indwelling urinary catheters only when necessary is the single most crucial step that can lower the prevalence of CAUTI [117]. Catheterization should only be used in acute circumstances, should be avoided, or used sparingly for the management of urine incontinence and chronic conditions, and alternatives to urethral catheterization should be researched. Several studies stress the significance of standardizing the insertion criteria for indwelling urinary catheters and, when necessary, reducing their usage entirely in favor of alternatives such as intermittent catheterization. Indwelling urinary catheters should be inserted using sterile procedures.

Enclosed drainage systems may lower the infection risk, but there is no concrete evidence that they reduce the frequency of CAUTI [118]. The drainage system must be emptied with the utmost care using aseptic methods, and the same collecting unit should never be used in any other patient.

Flores-Mireles et al. recommend utilizing the CDC “prevention guidelines” in bundles to prevent CAUTIs [119]. These guidelines and recommendations state that urinary catheterization should not be used to treat incontinence in patients or nursing home residents, that urinary catheters should only be used in surgical patients when absolutely necessary, and that the Foley catheter should be removed as soon as possible after surgery, preferably within 24 h [119]. This bundle provides effective ways to decrease CAUTI cases. Gray et al. reviewed the utilization of external urinary collection devices for males as a substitute to indwelling urinary catheters to reduce CAUTI [120]. The types of external collection devices include condom catheters, reusable body-worn urinals, and non-sheath, glans-adherent external collection devices [120]. Patients who are deemed to be particularly high risk for developing CAUTI can use anti-infective catheters [121].

Current research has demonstrated the efficacy of multimodal UTI prevention techniques and combinations in critical care units [122]. A specialized set of interventions for CAUTI prevention, education, outcome and process surveillance, feedback on CAUTI rates, and performance indices for infection control procedures are a few examples of such approaches. These techniques have been successfully used in critical care settings for both adults and children. Critical care units (CCUs) have seen a decrease in CAUTI rates thanks to these multifaceted infection control regimens, which were also linked to better hand cleanliness [121,123,124]. Hence, improvements in care practices can lower the likelihood of CAUTI and its negative effects, especially in CCUs in resource-constrained nations. Sustained, continuous improvements also must be made in community practices and extended care facilities.

CAUTIs are the most commonly acquired infection in hospitals probably because 15–25% of patients are catheterized during hospitalization [125]. The Institute for Health Care Improvement reports that 80% of all UTIs are caused by indwelling urethral catheters [126]. Increased costs, longer hospital stays, and increased mortality rates are some of the adverse impacts of CAUTI [125]. Hospitals are aware of the need to prevent CAUTIs due to their cost and the lack of reimbursement from CMS. Extended care facilities can prevent CAUTIs by implementing evidence-based practices, such as insertion criteria, providing proper perineal care, and timely removal of urinary catheters [125]. Use of silicone, instead of latex, catheters is shown to reduce the incidence of CAUTI [127].

## 6. Biofilms and Drug Resistance

The biofilm’s structural characteristics and constituent bacteria are responsible for the development of antibiotic resistance. Drug resistance associated with biofilms is a highly nuanced phenomenon that may largely be driven by biofilms. Antibiotics, disinfectants, and germicidal agents may be used to treat biofilms and associated infections. Bacteria that inhabit biofilms show a 10 to 1000-fold increase in drug resistance, especially antibiotic resistance when compared to their planktonic state [128]. For example, it has been reported that when examined planktonically, all *Staphylococcus epidermidis* biofilm isolates were sensitive to vancomycin [129]. However, when the same strains were isolated from a biofilm, approximately 75% were resistant to the same antibiotics [129]. Similar observations were made with *Klebsiella pneumoniae*. This strain is sensitive when isolated from an aqueous solution but extremely resistant to specific antibiotics when tested from a biofilm [130]. Delayed or inadequate diffusion of the anti-microbial agents through the biofilms, varied growth rate of the biofilm microbial organisms, and other functional alterations are responsible for the development of antimicrobial resistance. Bacteria in the planktonic state are known to use either enzymes, efflux pumps, or mutations as a means to evade drugs and develop resistance [131,132,133,134,135,136,137,138,139]. However, these studies also do not preclude the likelihood that conventional resistance mechanisms in biofilms play a role in antibiotic resistance. It was previously observed that *Pseudomonas aeruginosa* which was isolated from a biofilm that was repeatedly exposed to ceftazidime exhibited the conventional form of resistance to antibiotics [140]. Within a biofilm, a number of distinct components work together to mitigate or completely prevent the effectiveness of drugs, which further drives resistance. Strategies such as decreased penetration of the drug [141,142,143,144,145,146,147], the biofilm’s modified chemical microenvironment [148,149,150,151,152], and a subgroup of microorganisms in a biofilm that exhibit a form of differentiation similar to spore formation [153,154,155] all are linked to drug resistance in the biofilm colonies of bacteria [156]. This phenomenon, which is also referred to as recalcitrance, occurs when bacteria within the biofilm are able to survive in the face of high doses of drugs thanks to the combination of several different processes [157]. The maintenance of a high number of bacterial cells that survive antibiotic treatment due to tolerance of the slow-growing population and the presence of persisters [158,159], a high mutation rate, and the presence of antimicrobial-selective pressure, as well as localized competition in the compartmentalized structure of the biofilms between mutants, also contribute to the facilitation of the development of antibiotic resistance in biofilms. However, these studies also do not preclude the likelihood that conventional resistance mechanisms in biofilms play a role in antibiotic resistance. It was previously observed that *Pseudomonas aeruginosa* which was isolated from a biofilm that was repeatedly exposed to ceftazidime exhibited the conventional form of resistance to antibiotics [140]. Through the phenomenon of diffusion–reaction inhibition, EPS can inhibit the activity of antibiotics which disperse through biofilms [160,161]. This could lead the antibiotics in question to be chelated and form complexes, or its destruction through enzymatic breakdown [162,163]. The EPS part of the biofilm either slows the process of penetration or reacts with the antimicrobial agent and alters its effectiveness, further inducing resistance to treatment. Suci et al. showed *Pseudomonas aeruginosa* biofilms delayed penetration time from 40 to 21 min [164]. DuGuid et al. observed that the susceptibility of *S. epidermidis* to tobramycin remained diminished by its organization within the biofilm [165]. Hoyle et al. confirmed a 15-fold increased tobramycin susceptibility of dispersed bacteria compared to bacteria in intact biofilms [166]. Souli and Giamarellou demonstrated reduced susceptibility of Bacillus subtilis to many antimicrobial agents in *S. epidermidis* slime [167].

Another potential explanation for drug resistance is the slower growth rate of biofilm-associated bacteria. Slower growth results in slower uptake of antimicrobial agents that leads to suboptimal intracellular drug concentrations for bacterial killing. Anwar et al. showed that younger biofilms’ faster-growing cells were more liable to the antimicrobials than those in older biofilms [168]. Similarly, Tresse et al. demonstrated that *E. coli* in agar demonstrated more resistance to antimicrobials due to reduced oxygen tension [169]. A deeper understanding of biofilm has led to more effective novel therapies and tactics that operate as a barrier between bacteria and antimicrobial agents [170,171,172].

Yet another method of antibiotic resistance of biofilm microorganisms is the horizontal gene transfer, which is a process by which the bacteria acquires genes for resistance [128]. Moreover, a number of investigations have documented in vitro that mycobacterial biofilms were found to be resistant to either antibiotics (amikacin and clarithromycin) or disinfectants [173,174]. Several studies [90,91,92] that examined the impact of antibiotics at various stages of biofilm development found that antibiotic treatment was more successful at the beginning phases of biofilm development, likely as a result of the cells that have not yet fully adjusted to biofilm communities [175,176]. It has been proposed that the permeability of anti-tuberculosis medications was different among the mycobacterial species [174]. The development of antibiotic resistance in mycobacteria depends on the metabolic state and activation of resistance genes (such as methylases) [173,177].

## 7. Methods to Control Biofilms

Biofilm control is a difficult problem to solve. The frequency and severity of infections may grow, which would increase infection-related mortality and morbidity if biofilms are not properly combated [178]. As a result of the frequent use of antibiotics to treat biofilm-associated infections, more virulent, antibiotic-resistant bacteria have started to appear, necessitating the creation of cutting-edge techniques to eradicate them [179]. To effectively regulate biofilms, it is important to have a complete understanding of how they arise, adopt effective communication tactics, and put these strategies into action [180]. The designing of surfaces to limit bacterial adhesion, preoperative and postoperative precautions, and coating implants with antimicrobial medications are only a few strategies to avoid bacterial adherence [29,181]. Early biofilm formation can be stopped by antibacterial drugs, dietary supplements, surface treatments, and adjustments to other environmental factors [182,183]. Once established, a biofilm is difficult to get rid of, although it can be reduced and possibly even eliminated with the help of EPS antagonists, dissociation drivers, vaccination treatment, and mechanical elimination [184,185,186,187] (Figure 4). Several physical, biological, medicinal, and combinatorial techniques have been applied in clinical settings to dissolve mature biofilms [179,188].

At any point in the process of biofilm formation, it is possible to block it. Several strategies, such as the release of antimicrobial medications, preoperative and postoperative measures, anti-biofilms, and surface engineering, can be used to avoid bacterial adhesion. Use of various antibacterial agents, dietary supplements, anti-surface attachment treatments, and changes in ambient conditions can all prevent bacterial growth. Compared to an immature biofilm, it is more difficult to disrupt a developed biofilm. For developed biofilms several approaches have been used, including the activation of biofilm dissociation promoters, monoclonal antibody treatment, the application of peptide-based vaccines, and the eradication of biofilm microorganisms by standard cleaning processes. Physical approaches (including ultrasound, shear stress, and thermal shock therapy), biological (such as anti-pathogenic biofilm, and phages), chemical (including the use of enzymes, such as nucleases, proteases, and galactases), complementary strategies (such as enzymes and phage combinations, and disinfect and abrasives combinations), and protection from separation, are all used to degrade matured biofilm (e.g., use of environmental factors to dysregulate nutrients and QS system).

### 7.1. Traditional Antibiotics

The inadequacy of traditional antibiotic treatments has resulted in biofilm conditions around the globe worsening. Biofilms constantly change their EPS, rendering them resistant to different antibiotics. Treatments that do not work well include those given orally, such as β-lactams, quinolones, aminoglycosides, and macrolide antibiotics. Bacterial β-lactamases hydrolyze β-lactams, rendering them incapable of penetrating biofilms and ineffective [189]. Efflux pumps are responsible for removing quinolones, aminoglycosides, and macrolides from microorganisms [190]. Quinolones perform worse in the biofilm’s anaerobic environment [191]. Although biofilms are challenging to manage with conventional antibiotic therapy, they have demonstrated a certain type of drug susceptibility when appropriately implemented and in high quantities. Combination therapies, such as those using antibiotic adjuvants or topical treatments at high doses, have been employed [191]. When applied topically in greater quantities, cephalosporins, aminoglycosides, monobactams, polymyxins, tetracyclines, and glycoglycines have been shown to be effective against biofilms [192]. Inhaling antibiotics such as colistin, tobramycin, aztreonam, ciprofloxacin, levofloxacin, or gentamicin has proved effective in treating lung biofilm infections. It has been demonstrated that applying certain antibiotics directly to tubes or medical equipment, such as vancomycin, minocycline, linezolid, daptomycin, tigecycline, rifampicin, or cephalosporins, might reduce the growth of biofilms [191,193].

### 7.2. Replacements to Traditional Antibiotics

To tackle the problem of biofilm resistance to traditional antibiotics, novel approaches to the prevention and treatment of biofilm-associated infections are being investigated and developed. Next generation antibiofilm agents that are currently being evaluated include small molecules that can block particular virulence factors and specific matrix-targeting enzymes [194]. Studies on the biofilm of *S. epidermidis* have reported that DNase, proteinase K, and trypsin are some of the enzymes that target matrix proteins and break down eDNA, reducing the firmness and durability of the biofilm [183]. N-acetylcysteine and benzimidazole both are known to promote biofilm breakdown and prevent EPS generation [195,196]. Using specific chelators that trap divalent cations such as calcium and magnesium, compounds can decrease microbial adhesion and stop germs from clinging to material surfaces where biofilm formation is initiated [197]. Medical equipment surfaces are treated with bacteriostatic or bactericidal substances to prevent bacteria from attaching to them. For instance, vancomycin and other broad-spectrum antibiotics can be coated on metal implants to prevent the growth of biofilms on such surfaces. There is evidence for and against such an approach as studies have reported the inhibition of *S. epidermidis* biofilm growth and another study has reported that bacteria become resistant to vancomycin [198,199]. By destroying microbial proteins and DNA, anti-bactericidal chemicals such as silver, for example, may prevent the production of biofilms [200]. Although coating medical equipment with silver nanoparticles has in general demonstrated a positive effect, it is well known that excessive silver exposure is damaging to human cells [201]. There are also reports of agents being used to prevent surface attachment by some bacteria, such as Bacillus subtilis, by the use of bio-inspired quercetin nanoparticles [202]. Another antimicrobial compound employed as a coating, furanone, prevented *S. epidermidis* from forming biofilms [203]. However, the selection of coating agents is also based on the unique property of the surfaces on which they are being coated. A synthetic compound, Trimethoxysilyl propyldimethyloctadecyl ammonium chloride (QAS-30), is known to have bactericidal and bacteriostatic properties owing to its quaternary ammonium groups, and is typically used only to coat silicone-based devices, as they preferentially stick to silicone better [204]. Another method for preventing the growth of biofilms on medical equipment and prostheses is to use an anti-adhesion coating to repel bacteria off surfaces. In this method, the bacterial protein compatibility is decreased by the coating’s coarseness or charged surface [205]. Several examples of this method of coating are Trimethylsilane (TMS), selenocyanatodiacetic acid (SCAA), and polymer brush [17]. Biofilms, particularly those formed by *S. epidermidis*, were significantly inhibited by titanium and stainless-steel medical equipment, such as implants. An alternative novel method of coating is the use of an organo-selenium chemical called SCAA. In this method, the release of superoxide radicals by the compound is known to represses the growth of biofilm on coated surfaces of hemodialysis catheters [17,206]. Yet another type of anti-adhesion coating is the use of a polymer brush which involves the use of polyethylene oxide (PEO) and the generation of a repulsive osmotic pressure to ward off bacteria. However, this approach has been unsuccessful in vivo because of its weak adhesion to moist medical devices, where microbial flagella and pili increase movement and adhesion, and is more successful in vitro [17,207]. Although most opportunistic microbes that produce biofilms can infect people, some of them can shield people from more dangerous infections [208].

New antibiofilm compounds, which include halimane diterpenoids, imidazole and indole derivatives, plant extracts, peptides, and polysaccharides, are made of both natural and synthetic components [209]. Biofilms prevention strategies have been studied using degrading enzymes, drug delivery nanoparticles, and cell-damaging photodynamic therapy (PDT) [139]. Halimane diterpenoids, which are present in bacteria, marine creatures, and terrestrial plants, may be the most widespread naturally occurring antibiofilm agents. Due to their proven antibacterial and antimycobacterial properties, these adaptable compounds are employed industrially in agricultural, medicinal, and beauty products [210,211]. The biofilms produced by methicillin-resistant *S. aureus* (MRSA), *A. baumannii*, *P. aeruginosa*, *V. cholerae*, *K. pneumonia*, and *Shigella boydii*, have been prevented and eliminated using the natural imidazole derivative 2-aminoimidazole. By their impact on bacterial cell signaling and gene expression, indole derivatives have also been demonstrated to be effective against biofilms [209]. Plant extracts such as garlic, hordenine, limonoids, and quercetin are a few examples of natural products that have shown efficiency against biofilms by preventing the transcription of specific genes necessary for QS [212]. Studies have also reported the development of virulence factors inhibitors from cranberry polyphenols [213]. These virulence factors are necessary for bacterial adhesion and the bacterial structures and inhibition of these factors can be useful in preventing biofilm formation [214]. The human body produces LL-37, which is an anti-biofilm peptide that prevents bacteria from adhering to and creating biofilms on mucosal surfaces. In recent studies, *S. epidermidis*, and *P. aeruginosa* strains were reported to have been successfully treated with LL-37 [215]. Galactan and galactose, two larger polysaccharide molecules, are also known to negatively influence the development of biofilms. *Kingella kingae* produces galactan, which stops other bacterial species from developing biofilms [209,216]. A recent study has reported that in *P. aeruginosa* [217], ethylcholine, a newer form of choline, inhibited biofilm formation without affecting bacterial growth [217].

Given the formidable resistance of biofilms to currently available antibiotics, innovative approaches are being utilized to administer these conventional drugs. The resistance to conventional antibiotics is due to the poor penetration of these drugs across the biofilm EPS. Additionally, the hypoxic environment prevailing in the biofilm makes it difficult for antibiotics to spread across the biofilm making the biofilm mor resistant to conventional antibiotics. Another common technique is phage therapy, which prevents biofilm from forming by depolarizing EPS. Phage therapy can break up biofilms, although this happens over time. Although bacteria in biofilms can emit anti-phage compounds that neutralize phage, certain mature biofilms are resistant to phage therapy, thereby increasing resistance to this mode of therapy. Furthermore, the immune system quickly eliminates phage proteins from the body, decreasing its half-life and increasing resistance to such an approach. Phage therapy has another significant drawback in that it is not applicable to different types of biofilms because it is more tailored to specific bacterial strains. Biofilm disruption has been achieved using fresh, cutting-edge tactics. To manage and eliminate biofilms, ongoing research has focused on inhibitors of QS system (3), therapy with monoclonal antibody (4), and the use of natural ingredients (5). The effectiveness of new natural remedies in removing biofilm that is more resistant to conventional antibiotics is being tested. There are also reports, chloroform, andrographolide, ethanol extract of Houttuynia cordata poultice (HCP), suppressing *P. aeruginosa* and *S. aureus*-initiated biofilms by inhibiting several factors involved in bacterial survival and virulence.

### 7.3. Small Molecules

A wealth of knowledge gained in the mechanism of biofilm forming process has led to the creation of small molecules such as dimethylaminohexadecyl methacrylate (DMAHDM) composite that can inhibit the formation of biofilms. Dentists have researched this tactic to prevent the growth of *S. mutans* and *S. sobrinus*, which cause caries [218]. In other sectors, this method has been used to reduce the incidence of biofilm-induced infection. Nonetheless, the emergence of antibiotic-resistant bacteria has prompted a review of bacterial processes and new genetic discoveries to counter the threat caused by biofilms [185,219].

Small molecules are compounds with a molecular weight of less than 1000 Da [220]. They are preferred alternatives to standard antibiotics for the control of biofilms due to unique properties, such as high cell permeability, excellent stability, low cost, and low toxicity [221,222]. Over the course of the last few decades, numerous natural and synthetic small molecules with excellent antimicrobial properties have been identified. In silico screening is the method of choice for identification and the methodology relies on molecular docking and dynamic simulation using phenotypic similarity as an important criterion for screening. The source databases for screening are different small-molecule libraries. Initial screening generates a list of small molecules that can interfere with a particular function of microbial growth, accompanied by a range of scores and probabilities [223]. Identifying small molecules from this database is an automated process; it is quick and identifies lead small molecules with high probability for success. Small molecules identified by the silico method then need to be screened in vitro for their validation. The most widely used databases for screening include ZINC, PubChem, ChemSpider, DrugBank, and MCE. These databases store information pertaining to bioinformatics, cheminformatics, and specific targets of various small molecules [224].

Using such screening techniques, small molecules have been identified and validated against *S. mutans. S. mutans* is able to colonize the surface of the tooth and create biofilms, which help increase its virulence and shield the bacteria from the effects of antibiotic treatment [225]. For *S. mutans* biofilms to form, key components such as antigens I and II, glucosyltransferases (Gtfs), sortase A (SrtA), and quorum-sensing (QS) systems are required [226,227,228]. ZINC19835187 (ZI-187), ZINC 19924906 (ZI-906) and ZINC19924939 (ZI-939) are three molecules identified by Rivera-Quiroga and colleagues after screening a total of 883,551 molecules from the “Small molecule” library. These chemicals block *S. mutans* adhesion to polystyrene microplates by targeting antigens I/II [229]. Using a similar screening strategy, a small molecule was discovered by Chen et al. after screening the small molecule library of oxazole derivatives. The investigators identified 5H6[2-(4-chlorophenyl)-4-[(6-methyl-2-pyridinyl)amino]methylene-1],3-oxazole-5(4H)-1, which prevents the synthesis of EPS and *S. mutans* biofilms by inhibiting GtfC and GtfB. Additionally, a single active compound, 2A4, was identified by Wu et al. from a small-molecule library of 506 compounds. 2A4 selectively inhibits *S. mutans* in multispecies biofilms and inhibits both planktonic cells and single-species biofilms by reducing gene expression of antigens I/II and Gtfs [230]. Wu et al. discovered the lead chemical G43 through a structure-based virtual screening of 500,000 compounds against the GtfC catalytic domain [231]. G43 preferentially links GtfC and prevents *S. mutans* from forming biofilms and becoming cariogenic [231]. Furthermore, Ren et al. screened 15,000 molecules based on the structure of the *S. mutans* GtfC protein domain. They discovered a quinoxaline derivative called 2-(4-methoxyphenyl)-N-(3-[2-(4-methoxyphenyl)ethyl]imino-1,4-dihydro-2-quinoxalinylidene)ethanamine. This quinoxaline derivative selectively binds to GtfC and decreases the synthesis of insoluble glucans and biofilms formed by *S. mutans*, which in turn prevents the formation of caries in living organisms [232]. SrtA is a membrane-bound transpeptidase that plays a role in the biofilm formation of *S. mutans* by anchoring antigens I/II to the cell wall. This action is necessary for the creation of the biofilm. To find a SrtA inhibitor, Samanli et al. screened 178 small compounds from a library and selected CHEMBL243796 (kurarinone), which has a higher binding affinity with SrtA than CHX and displays improved pharmacokinetic activity toward *S. mutans* [233]. The ZINC library and the TONGTIAN library were searched by Luo et al. who identified numerous possible inhibitors of SrtA [234]. These prospective inhibitors include benzofuran, thiadiazole, and pyrrole, all of which bind to SrtA and reduce its activity. These SrtA inhibitors have shown to have potential for the management of *S. mutans* biofilms [234]. Using comparable screening strategies, Ishii et al. tested 164,514 small molecules against the peptidase domain of ComA, a critical component of *S. mutans* QS, and found six compounds that reduce biofilm formation without inhibiting cell growth [235].

Another significant phenotypic characteristic connected to *S. mutans’* cariogenicity is acid tolerance [236]. An essential enzyme for S. mutans’ ability to overcome acidity is the proton pump F1F0-ATPase (H^+^-ATPase) [237]. Sekiya et al. used the small molecule library to screen inhibitors against F1F0-ATPase and identified piceatannol, curcumin, and desmethoxycurcumin (DMC; a curcumin analog) as potential inhibitors of F1F0-ATPase, which can further result in decreasing growth and survival of *S mutans* in acidic conditions suggesting a potential anticaries strategy by inhibiting F1F0-ATPase, according to [238].

### 7.4. QS System Blockers

QS is a very unique microbial communication system based on unique biochemicals produced by the microbes that control the generation of virulence factors and the development of biofilms, among other biological processes [239]. Therefore, inhibition of genes and protein factors involved in imparting virulence can be successfully inhibited by QS system inhibitors [240]. Experiments with polypeptides [241], cephalosporins [242], aminoglycosides [243], and quinolones [244] showed that QS-inhibitors function synergistically to prevent biofilm development, increasing the efficiency of some medications. Using anti-QS chemicals that are readily available on the market, research by Brackman et al. has shown that bacterium biofilms are more vulnerable to antibiotics both in vivo and in vitro [216]. Therefore, use of antibiotics in conjunction with QS system inhibitors is being suggested as an alternative to conventional antibiotics. QS-inhibitors, such as N-(4-[4-fluoroanilno]butanoyl)-L-homoserine lactone (FABHL) and N-(4-[4-chlororoanilno]butanoyl)-L-homoserine lactone (CABHL) are known to prevent the QS system of *Pseudomonas aeruginosa* by down-regulating the expression levels of lasR and rhlR genes [245]. Studies using polypeptides [241], cephalosporins [242], aminoglycosides [243], and quinolones [244] showed that QS-inhibitors function cooperatively to prevent biofilm development, increasing the efficiency of some medications.

### 7.5. Therapy with Monoclonal Antibodies

Although antibody-based antibiofilm therapy has shown promise in preclinical models that target a number of biofilm components, its application in vivo has been constrained by poor target specificity and infusion responses [246]. Studies have shown that treatment with a pool of monoclonal antibodies has reduced biofilm production and avoided infection caused by biofilms. For instance, when utilized against *S. epidermidis*, monoclonal antibodies 12C6, 12A1, and 3C1 showed growth suppression and attachment to the bacterial accumulation-associated protein (AAP) [246]. The native human monoclonal antibody TRL1068 binds to DNABII proteins from both Gram-positive and Gram-negative bacteria, disrupting the biofilms of *S. aureus* and *P. aeruginosa*. When combined with antibiotics, TRL1068 inhibits the growth of biofilms [247]. To prevent the growth of biofilms, monoclonal antibodies in particular target antigens such adhesin proteins (ClFA, FnBPA, Can, and SasG). To prevent EPS dynamic changes, they bind to and deactivate cell wall-modifying enzymes (Atl, Atl-Amd, Atl-Gmd, and IsaA) [248,249,250,251,252]. The promotion of opsonophagocytic killing (OPK) effects by monoclonal antibodies against glycopolymers (WTA, CP, and LTA) has also been reported [252,253]. Studies have also been performed using anti-matrix component antibodies such as PNAG and DNABII, immune evasion proteins such as Spa, toxins such as HIa and LukAB, and proteins such as PhnD, showing positive results [254]. 

### 7.6. Inhibitors of Biofilm from Natural Resources

Clinical practitioners and researchers working on antibacterial therapeutics are very concerned about the formation of biofilms by multi-drug resistant microorganisms. To counter this, a number of bioactive substances with antimicrobial properties have been isolated from natural resources, purified, and put through clinical trials to see how well they inhibit biofilms [254]. Such natural products target bacterial cells in multiple ways. Product extracts using chloroform and ethanol have been demonstrated to prevent biofilm formation. Such prevention properties have been shown by andrographolide, sodium houttuyfonate, and emodin (Figure 4). The chloroform extract of Andrographis paniculata exhibited a reduction in the extracellular virulence factors controlled by quorum sensing (QS) in *P. aeruginosa* infections. Additionally, it impeded migrating motility and biofilm formation by downregulating the expression levels of p38 and ERK in the MAPK signal pathway [255]. Andrographolide is known to have efficacy against *E. coli* by inhibiting fimA and pap (TSH), and in recent research, it was demonstrated that it destroyed the biofilm of *S. aureus* by blocking the transcriptional factor SarA [256,257]. Anaerobic biofilms have been controlled and removed using natural product extracts in ethanol, such as in the reduction of *P. gingivalis* through the inhibition of IL-8 and CCL [258,259]. For a long time now, sodium houttuyfonate has been used in clinical practice as an antimicrobial agent. It is derived from the oil of the plant *Houttuynia cordata* and has been demonstrated to reduce the transcriptional levels of autolysis, specifically cidA in *S. aureus* and *S. pneumoniae*, thereby preventing the growth of biofilms by these species of bacteria [260]. It has also been shown that natural medications such as emodin and ajoene can regulate biofilms formed by a number of bacteria, including *P. aeruginosa*, *E. coli*, and *S. aureus* [261,262,263,264].

### 7.7. Probiotics

Live bacteria known as probiotics provide the host with health benefits when given in sufficient quantities. Probiotics are typically microorganisms that have inhibitory activity against specific pathogens, adhesion to epithelial cells, and resistance to specific concentrations of bile and acid [265,266]. Moreover, they must be resistant to antibiotics and be able to bind to pathogens tightly enough to render them inactive [267]. Antibiotic resistance has drawn attention to probiotics that prevent the production of biofilms. *E. faecium* WB2000, *Bifidobacterium BB12*, and *Bifidobacterium adolescentis* SPM1005 are probiotics that seem to prevent the development of biofilms [268,269,270,271]. Probiotics offer a variety of chemicals that assist in preventing colonization [271,272], decrease bacterial adhesion [272], enhance the immune system [273], release bacteriocin to hinder bacterial growth [274], maintain lactic acid in the environment to diminish bacterial virulence signaling systems [275], suppress EPS from pathogens [276], and increase indole in biofilm to down-regulate the QS system [277] (Figure 5). By creating pores in the cell walls of pathogens, probiotic byproducts such as bacteriocin can lyse them [278]. The biofilm’s pathogenic microcolonies may also be harmed by hydrogen peroxide (H_2_O_2_) [279]. Probiotics might be essential in preventing the development and dispersal of pathogenic biofilm from the time of initial attachment and throughout the entire progression. This happens via a number of mechanisms, including anti-adhesion activities, QS-system suppressors, and the development of non-infectious biofilms that are able to compete with nutrients and pH changes [280,281,282].

### 7.8. Using Gene Editing Methods

The application of genetic alterations techniques is a novel way to reduce virulence of pathogenic bacteria. By utilizing technologies for gene editing such as Clustered Regularly Interspaced Short Palindromic Repeats (CRISPR)-associated (CRISPR-Cas) systems, it may be feasible to change the genetic makeup of biofilm pathogens, which over time may reduce their virulence [283,284]. By focusing on the right genes, CRISPR-Cas has been utilized to modify the genetic makeup of bacteria and reverse antibiotic resistance [285,286]. This method has also been used to reduce the resistance to different antibacterial medications brought by the plasmids of selected infectious bacteria, such *E. coli*, and to focus on particular genes involved in virulence and antibiotic resistance in bacterial populations [287,288,289]. Studies have demonstrated that antibiotic resistance-causing bacterial genes can be neutralized using the CRISPR-Cas system [288,290]. Additionally, gene editing methods are being developed to target virulence genes and distribute polymeric nanoparticles, either alone or in combination with other delivery mechanisms as conjugative delivery or phage delivery [291]. Recent investigations on gene editing approaches for anti-biofilm purposes include those by Zuberi et al. to prevent the growth of *E. coli* biofilms [292], Tang et al., to prevent the growth of *S. mutans* biofilms [293], and Garrido et al., to destroy *S. aureus* biofilms in vivo [294].

### 7.9. Anti-Infective Devices

The crucial phases in the growth of medical device-associated bacterial infections include bacterial adhesion and ensuing biofilm formation on the surface of medical devices. Medical equipment surfaces, especially hydrophobic surfaces, are particularly susceptible to planktonic bacteria adhering to them. The addition of antibiotics or other chemicals to the drainage bag, the external application of antibiotic compounds, the use of certain meatal cleansing agents, and the coating of catheters with antibiotics or hydrogels are different methods that have been tried without success. Examples of this approach include polyvinylpyrrolidone (PVP), salicylic acid-releasing polyurethane acrylate polymers, antimicrobial peptides conjugated to co-polymer brushes, nanoparticles, hyaluronic acid coating, low-energy surface acoustic waves, and anti-adhesion agents. Therefore, new antimicrobial drugs or agents that inhibit bacterial virulence and aggregation, and biofilm formation or dissolution are needed.

Both antiseptic- and antimicrobial-coated urinary catheters, which have both been researched for the prevention of catheter-associated bacteriuria, are options for anti-infective catheters. Silver alloy is used to coat antiseptic-coated catheters currently on the market, since silver oxide coatings could not prevent CAUTI [164]. Catheters that have an antimicrobial coating include those coated with nitrofurazone, rifampicin, or minocycline.

A medicinal fungus called *G. lucidum’s* ethanol extract was used to create multifaceted silver nanoparticles with biomedical applications [295]. For catheterizations lasting fewer than 7 days, a meta-analysis to evaluate the advantages of silver-alloy-coated catheters showed a reduced risk of catheter-associated asymptomatic bacteriuria compared to conventional latex catheters [296]. Moreover, in individuals catheterized for fewer than 7 days, asymptomatic bacteriuria was less common when using antimicrobial-coated catheters [121]. 

To lessen the risk of bacterial colonization, catheters have been designed that are coated with antiseptic and antibacterial substances, such as silver ions, antibiotics, and noble metal alloys (NMA) [296]. In numerous contexts, it has been demonstrated that an NMA-coated latex catheter with a non-releasing coating of gold, silver, and palladium lowers the frequency of CAUTI (e.g., ICU, burn units, rehabilitation). Utilizing anti-infective catheters has not been conclusively linked to the prevention of CAUTI, the occurrence of bacteremia secondary to urosepsis, or reduced mortality, despite various research studies that show a decreased incidence of asymptomatic bacteriuria. Anti-infective urinary catheters maybe be offered to individuals who are thought to be at an especially high risk of developing CAUTI, or in cases in which other preventive measures have failed to lower CAUTI rates in a CCU. Systemic antibiotics are not advised for the prevention of CAUTI [297]. As a result, research on CAUTI prevention is still ongoing, and CCU practices should constantly take the most recent UTI recommendations into consideration.

## 8. A Prospectus for Future Research

Biofilm infections continue to be a major concern in the healthcare industry due to their high level of resistance to available antimicrobial medicines. Given the resistance to currently utilized antimicrobial drugs, it is vital to find effective treatment options for biofilm-associated infections. There are a few novel and successful antibiofilm techniques that have been tested, including isolation of quorum-quenching compounds, dispersal of produced biofilms, and combining antibiotics with quorum-quenching compounds. Although the aforementioned strategies are significant areas of research, they are still in their infancy and have not yet completed clinical trials or made their way into the commercial market. Future research on strategies to prevent and manage biofilm infection should focus on different preventative and remedial measures against biofilm colonization of medical devices. Additionally, it should involve studies aimed at combating antimicrobial resistance. The ubiquity of the biofilm phenotype has been acknowledged by the discipline of microbiology. To better understand microbiologic processes, researchers in the domains of medical, dietary, water, and environmental microbiology should focus on the danger of biofilms. The pharmaceutical and healthcare sectors’ adoption of this strategy will surely lead to the development of new biofilm prevention and control techniques. Understanding what distinguishes the biofilm phenotype from the planktonic phenotype in greater detail will hold the key to success for future measures to control biofilms.

## Figures and Tables

**Figure 1 microorganisms-11-01614-f001:**
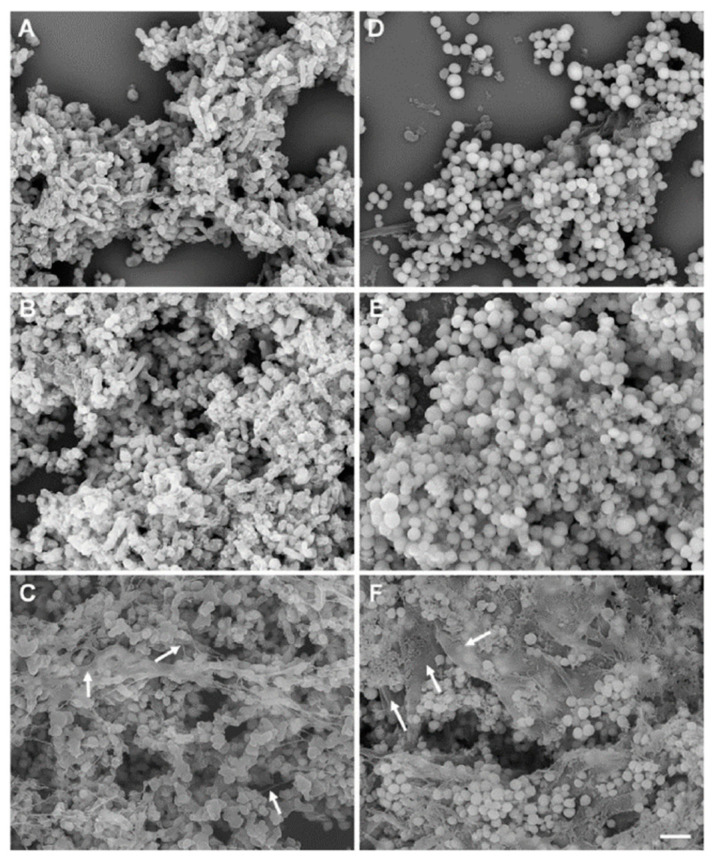
A conventional scanning electron microscopic (SEM) image of a biofilm formed by *M*. *haemolytica* (D153), grown in colorless RPMI 1640 (**A**–**C**) and *S. aureus* Newbould 305 (NB305), grown in BHI broth (**D**–**F**) on round glass coverslips in 24-well plates at 37 °C. Biofilms grown on glass coverslips were fixed with 10% formalin (**A**,**D**), 2.5% glutaraldehyde (**B**,**E**), or Methacarn (**C**,**F**) fixative solutions for 48 h and samples were further processed for SEM examination. EPS layers on the top, in the middle, and in the bottom of biofilms (**C**,**F**) are shown by white arrows. This figure is reproduced from PLoS One. 2020; 15(5): e0233973. This is an open access article, free of all copyright, and is reproduced under the Creative Commons CC0 public domain dedication.

**Figure 2 microorganisms-11-01614-f002:**
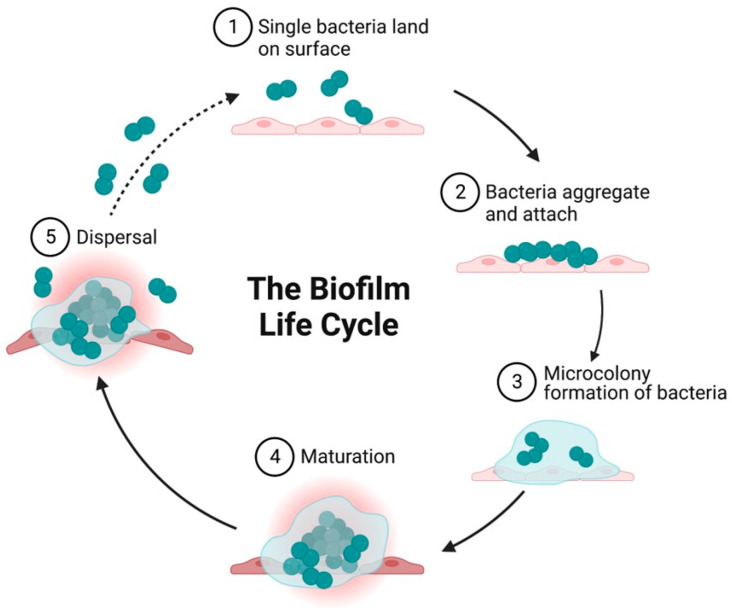
Diagrammatic illustration showing the growth cycle of a biofilm by a single bacterium species on a solid surface. (1) Reversible attachment of single planktonic bacteria to surfaces. The first attachment of the bacteria is influenced by attractive or repelling forces generated by nutrient levels, pH, and the temperature of the surface. (2) Aggregation of bacteria and irreversible attachment to surfaces. (3) Formation of an external matrix of multilayered complex biomolecules, microcolony formation, and EPS secretion that constitute the external matrix. Secretion of polysaccharides in biofilm forming strains enables aggregation, adherence, and surface tolerance, allowing for improved surface colonization. (4) Maturation of biofilms and acquisition of a three-dimensional structure as they reach maturity. These three-dimensional structures rest on self-produced extracellular matrix components. (5) Fully mature biofilms detach, which allows bacterial cells to take on a planktonic state once again and thereby establish biofilm in other locations. Created on BioRender.com 31 March 2023.

**Figure 3 microorganisms-11-01614-f003:**
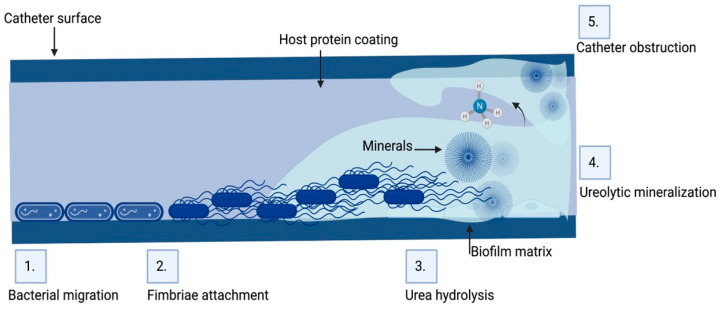
Biofilm formation and pathogenesis mechanism of CAUTI: The environmental conditions created on the catheter surface make it an ideal site for bacterial attachment and formation of biofilm structures. (1) Bacteria migrates through the periurethral area along the catheter surface. (2) Fimbriae attach to the body-fluid-derived catheter surface or directly to the catheter material inducing EPS production and biofilm formation. (3) Some bacteria such as *P. mirabilis* produce enzymes involved in the hydrolysis of urea in urine into ammonia, increasing the local pH leading to the production of minerals in urine which results in struvite crystals. (4) Struvite formed is incorporated into the developing biofilm—a process called ureolytic mineralization, which is also facilitated by the capsule polysaccharides. (5) Fully developed crystalline biofilm eventually causes catheter obstruction. Created on Biorender.com (31 March 2023).

**Figure 4 microorganisms-11-01614-f004:**
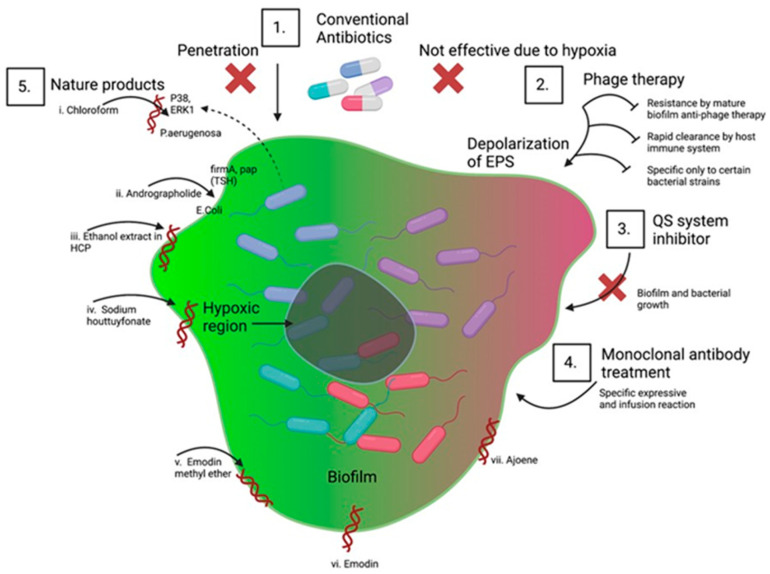
Conventional and novel methods to control biofilm: To fight infections ensuing from biofilms, numerous methods have been developed from different aspects; some are conventional, and some are novel. (1) Use of conventional antibiotics in the early stage; however, this method has a high failure rate due to poor penetration and lack of action due to hypoxia. (2) Phage therapy has been used as an alternate approach for controlling biofilm formation. This method works by depolarizing the EPS to disrupt the biofilm. This method also has multiple limitations including resistance, clearance by the host immune system, and specific only to certain strains of bacteria. (3) Novel methods of biofilm disruption such as QS system inhibitors which interfere with the microbial communication mechanism based on molecular signatures are also being used. (4) Newer methods such as antibody-based therapy against biofilm are being tried in preclinical models that target several biofilms but are limited in their success due to poor target specificity and infusion reaction. (5) Natural product-based therapy is another conventional method used. Products used here are either crude extract or purified compounds. Biologically active compounds showing antibacterial activity are extracted, purified, and successfully evaluated to clinical and pre-clinical models. Such extracts include chloroform, ethanol, and methyl ester. These extracts inhibit biofilm through various mechanisms ranging from inhibition of a critical enzyme involved in the growth of bacteria to repression of gene expression of multiple genes required for bacterial growth and development.

**Figure 5 microorganisms-11-01614-f005:**
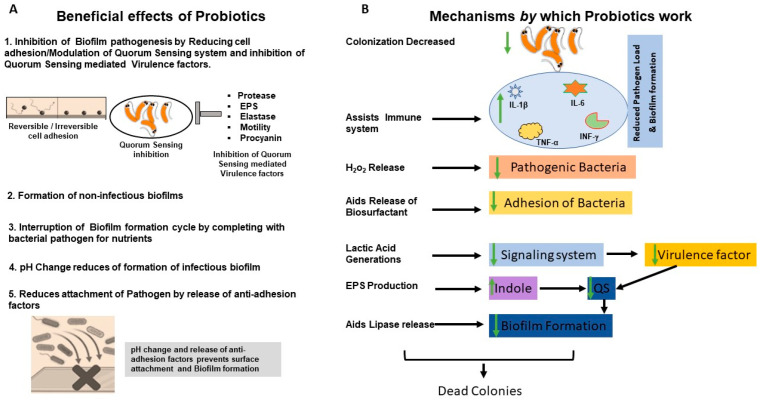
Probiotic methods to control biofilms. Probiotics pathogenic biofilm inhibition in different ways, as shown in the Figure. (**A**) Beneficial effects of probiotics (**B**) Mechanism of action of probiotics.

## Data Availability

Not applicable.

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
