# Peer review of "Microbial Biofilm: A Review on Formation, Infection, Antibiotic Resistance, Control Measures, and Innovative Treatment"

_microorganisms, 2023, doi:10.3390/microorganisms11061614_

Round 1

Reviewer 2 Report

In this manuscript a review of biofilm formation, its characteristics and control methods is made.

The following comments are made:

1. Line 28: Flora is no longer used to put microbiota. Correct in all text

2. Line 74. Italicize S. mutans. Correct the name of the bacteria in italics throughout the text, there are many errors throughout the text.

3. Line 84. Capitalizing Gram is a proper noun. Correct all over the text.

4. Line 105. Remove a comma

5. Line 122. Put what QS means

6. Line 158. Adhesion molecules and cell receptors were missing

7. Line 325. It is albicans. Correct

8. Line 338. Are they CFU? put the units

9. Line 390. Put what CAUTI means

10. Line 416. Put the name of the bacteria in italics. Correct all over the text.

11. Lines 563-611. Name bacteria in italics

12. Lines 649-656. What molecules? put the names

Put the names of bacteria in italics

Author Response

Reviewer 2, comment 1:  Line 28: Flora is no longer used  correct to .microbiota. Correct in all text.

Response: The authors thank the reviewer for pointing this out. The word flora has been changed to microbiota in line 28 and throughout the text. Since new text has been added to the revised document line 28 has moved to line 33. The other places where the word flora has been changed are lines- 382, 398, 441, and 446 in the revised document.

Reviewer 2 comment 2:  Line 74. Italicize S. mutans. Correct the name of the bacteria in italics throughout the text, there are many errors throughout the text.

Response: S. Mutants have been italicized throughout the text. However, line 74 is changed in the revised document because of new legend to figure section does not have old wordings.

Reviewer 2 comment 3:  Line 84. Capitalizing Gram is a proper noun. Correct all over the text.

Response: The authors thank the reviewer for pointing this out. The word gram has been capitalized on line 84 which is  line 97 in the revised document. Additional places where the word gram has been capitalized are lines 89,97,120,438, 852, and 853.

Reviewer 2 comment 4:  Line 105. Remove a comma.

Response: The comma has been removed and in the revised document the line number is 118.

Reviewer 2 comment 5: Line 122. Put what QS means.

Response: what QS means is added in the text and the new line number in the revised document is 135.

Reviewer 2 comment 6:   Line 158. Adhesion molecules and cell receptors were missing.

Response: Adhesions and microcolony formation 2 new sections have been added. In the revised document these two sections are from line 182 to line 219.

Reviewer 2 comment 7: Line 325. It is albicans. Correct

Response: The authors thank the reviewer for pointing out this typo. This is now corrected in the revised document and the new line number is 378.

Reviewer 2 comment 8: Line 390. Put what CAUTI means.

Response:  CAUTI is now expanded  in the revised document.

Reviewer 2 comment 9:  Line 416. Put the name of the bacteria in italics. Correct all over the text.

Response: The name of the bacteria has been italicized and the same has been done throughout the text in the revised document.

Reviewer 2 comment 10: Lines 563-611. name of bacteria in italics

Response: The bacterial name has been italicized and the same has been done throughout the text in the revised document.

Reviewer 2 comment 11: Lines 649-656. What molecules? put the names

Response: The name of the small molecules has been added and additional details of other small molecules have been added in the revised document lines 754-823

Reviewer 3 Report

This manuscript briefly summarizes the composition of biofilm, the types and properties of bacteria in biofilm, the formation of biofilm, the infection caused by biofilm, biofilm on medical devices, biofilm and drug resistance, and several methods to control biofilm infection.The article has a good idea, but there are several problems that need attention.

1 Why is the number 10 directly after the number 8.9?

2 We think this article focuses on the biofilm formation and treatment measures in medical devices, as mentioned in the abstract, so is the description of this part comprehensive?

3 In section 8, the author summarizes several methods to control biofilm, and we think it needs more detailed description, such as traditional antibiotics and their substitutes, which can be compared and expounded by tables and other methods.

4 As far as we know, biofilm has not only its harmful side, but also its beneficial side. Should it be expounded in this paper?

Author Response

Reviewer 3 comment 1: 1 Why is the number 10 directly after the number 8.9?

Response: This was an error; we thank the reviewer for pointing out that this now stands corrected in the revised document. Because of Merger of earlier sections as suggested by the reviewer this section is now 8 in the revised document.

Reviewer 3 comment 2: We think this article focuses on the biofilm formation and treatment measures in medical devices, as mentioned in the abstract, so is the description of this part comprehensive.

Response: We believe the description is comprehensive and for treatment measures now we have added further details on small molecules (lines 754-853) and the drug resistance (lines 522-591) section in the revised document. We hope this addresses the concern of the reviewer.

Reviewer 3 comment 3:  In section 8, the author summarizes several methods to control biofilm, and we think it needs a more detailed description, such as traditional antibiotics and their substitutes, which can be compared and expounded by tables and other methods.

Response: We appreciate the reviewer’s comment but believe we have added sufficient details of the traditional antibiotics and added more details on the section with respect to drug resistance. Figure 4  summarizes the different treatment approaches  which can be compared in the figure.

Reviewer 3 comment 4: 4 As far as we know, biofilm has not only its harmful side but also its beneficial side. Should it be expounded in this paper?

Response: We gave this question serious thought and realized that the beneficial effects of biofilms are in the field of agricultural and other industrial settings. Mainly in the area of phytopathogen and biofertilizer areas. Addressing the beneficial effects in a focused article on the Health and Medical effects of biofilms will be a deviation and  beyond the scope and aim of the article. Therefore, we respectfully disagree with the reviewer on this issue.

Reviewer 4 Report

pdf file

Author Response

Reviewer 4 comment 8:   Line 166: van der Waals ?

Response: the term van der Waals has been corrected

Reviewer 4 comment: Section 6: . Biofilms on medical devices – this section is not objectionable.

Response: This section has no changes as suggested by the reviewer.

Reviewer 4 comment 9:   Section 6.7 and 6.8 – this should be included in 6.6 or specified in some other way.

Response: We agree with the reviewer and the two sections have now been included in section 6.6 in the revised document.

Reviewer 4 comment 7: Section 7: Biofilms and drug resistance – this section should be described in more detail.

Response: More details have been added for this section Lines 522-592 have been added.

Reviewer 4 comment 8:   Section 10: A Prospectus for Future Research – it seems to me that some introductory, more general sentence is missing in this summary.

Response: We thank the reviewer for pointing this. We have now introduced new general statements in this section, lines 967-975. 

Round 2

Reviewer 2 Report

1.Italicize scientific names. Review throughout the text. Lines 438, 667, 715, 872, 892, 946.

2. Remove comma in line 117

The quality improved

Author Response

The reviewer  wanted us to

1.Italicize scientific names. Review throughout the text. Lines 438, 667, 715, 872, 892, 946.

  1. Remove comma in line 117
  2. The concern of the reviewer has been addressed all over the text
  3. Also in page the author afilation has been updated